# On Unsupervised Prompt Learning for Classification with Black-box Language Models

## Abstract

*Large language models* (LLMs) have achieved impressive success in text-formatted learning problems, and most popular LLMs have been deployed in a black-box fashion. Meanwhile, fine-tuning is usually necessary for a specific downstream task to obtain better performance, and this functionality is provided by the owners of the black-box LLMs. To fine-tune a black-box LLM, *labeled* data are always required to adjust the model parameters. However, in many real-world applications, LLMs can label textual datasets with even better quality than skilled human annotators, motivating us to explore the possibility of fine-tuning black-box LLMs with unlabeled data. In this paper, we propose *unsupervised prompt learning* for classification with black-box LLMs, where the learning parameters are the prompt itself and the pseudo labels of unlabeled data. Specifically, the prompt is modeled as a sequence of discrete tokens, and every token has its own to-be-learned categorical distribution. On the other hand, for learning the pseudo labels, we are the first to consider the *in-context learning* (ICL) capabilities of LLMs: we first identify reliable pseudo-labeled data using the LLM, and then assign pseudo labels to other unlabeled data based on the prompt, allowing the pseudo-labeled data to serve as *in-context demonstrations* alongside the prompt. Those in-context demonstrations matter: previously, they are involved when the prompt is used for prediction while they are not involved when the prompt is trained; thus, taking them into account during training makes the prompt-learning and prompt-using stages more consistent. Experiments on benchmark datasets show the effectiveness of our proposed algorithm. After unsupervised prompt learning, we can use the pseudo-labeled dataset for further fine-tuning by the owners of the black-box LLMs.

## 1 Introduction

Large language models (LLMs) have shown impressive performance in various text-formatted few-shot classification tasks, where the model takes the input text and outputs a corresponding label (Brown et al., 2020; Ouyang et al., 2022). In recent years, most popular LLMs have continued to grow in size and are increasingly deployed as black-box models, accessible through commercial application programming interfaces, such as GPT-3 (Brown et al., 2020). These black-box models provide promising results for data labeling and require minimal initial investment, as users can deploy them directly for downstream tasks. Due to their strong performance and low initial investment, the use of black-box LLMs for classification is gaining popularity in real-world applications.

The performance of LLMs in downstream tasks can be further enhanced by supervised fine-tuning algorithms (Houlsby et al., 2019; Hu et al., 2021). These methods always rely on *labeled* data for downstream tasks, allowing owners of black-box LLMs to fine-tune the model with these labeled data and deliver a customized version of the black-box LLM after fine-tuning[1]. In scenarios where fine-tuning the black-box model is not possible, the learner can employ prompt learning algorithms designed for black-box LLMs (Sun et al., 2022; Diao et al., 2022) to learn a prompt to label the specific downstream task, as both the model parameters and the prompt can influence the final performance. Nevertheless, these algorithms still require labeled data to learn the prompt.

With the continuing advances and successes in LLM research, most popular LLMs can now label textual datasets in many real-world tasks with quality equal to or better than skilled human annota-

---

[1]https://platform.openai.com/docs/guides/fine-tuning

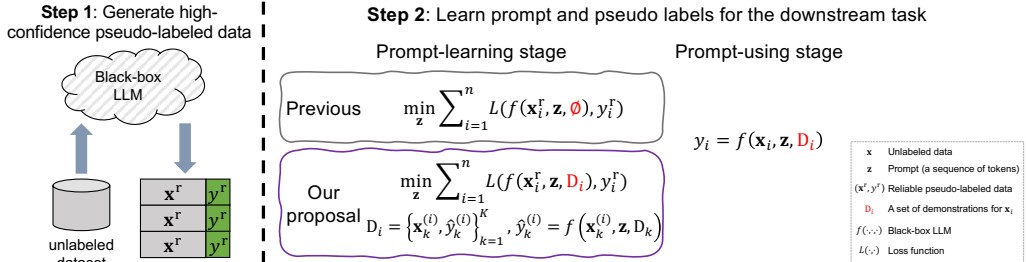

**Figure 1:** An illustration of our main idea. We first identify reliable pseudo-labeled data using the LLM and then learn pseudo labels on other unlabeled data together with the prompt by fine-tuning on these reliable pseudo-labeled data. In-context demonstrations are a crucial component of our proposal: previously, they are involved when the prompt is used for prediction while they are not involved when the prompt is trained; thus, taking them into account during training makes the prompt-learning and prompt-using stages more consistent.

tors (Gilardi et al., 2023; Törnberg, 2023). For example, ChatGPT has been shown to outperform crowd workers in text annotation tasks (Gilardi et al., 2023). Additionally, LLMs were found to be inherently highly capable of identifying label errors in natural language datasets (Chong et al., 2022). ChatGPT-4, in particular, outperforms both experts and crowd workers in annotating political Twitter messages using zero-shot learning (Törnberg, 2023). These findings motivate us to explore the possibility of fine-tuning black-box LLMs using unlabeled data, as the LLM can generate high-quality labels for fine-tuning, eliminating the need for costly human annotation. Our goal is to obtain accurate pseudo labels for the downstream task and then fine-tune the black-box LLM. A feasible method involves first assigning pseudo labels to the unlabeled data, identifying the reliable pseudo-labeled data, and then leveraging commercial fine-tuning services with these selected data. This process can be repeated iteratively, with each iteration adding newly identified reliable pseudo-labeled data, followed by further fine-tuning of the model by using the commercial fine-tuning services. However, this iterative process is quite complex, and relying solely on a limited number of reliable pseudo-labeled data at the beginning of the learning process carries a significant risk of overfitting.

In this paper, we propose *unsupervised prompt learning for classification with black-box LLMs*, where we simultaneously learn the prompt itself and the pseudo labels of unlabeled data. The learned pseudo-labeled dataset can then be used to further fine-tune the black-box LLM using commercial fine-tuning services. Inspired by recent discrete prompt learning algorithms for black-box LLMs (Deng et al., 2022; Diao et al., 2022), we assume the prompt is a sequence of discrete tokens, where each token is sampled from a categorical distribution over the entire vocabulary. We optimize these categorical distributions throughout the learning process. During inference, we sample each token according to its categorical distribution to form the prompt. On the other hand, for learning the pseudo labels, we propose to exploit the *in-context learning* (ICL) capability of LLMs (Brown et al., 2020; Wei et al., 2023). We use the pseudo-labeled data as in-context demonstrations to facilitate the simultaneous learning of pseudo labels and the prompt in the prompt-learning phase. As shown in Figure 1, we first identify reliable pseudo-labeled data using the LLM, and then learn pseudo labels on other unlabeled data together with the prompt by fine-tuning on these reliable pseudo-labeled data. Different from previous prompt learning methods that learn prompts based on labeled data and then use labeled data as demonstrations only in the prediction stage, we learn prompts and the pseudo labels simultaneously in the training process. The demonstrations are selected by a specific algorithm and assigned pseudo labels predicted by the LLM based on the prompt and their corresponding demonstrations.

These in-context demonstrations are important: they are previously only involved when the prompt is used for prediction, but they are not involved when the prompt is trained; thus, taking them into account during training makes the prompt-learning and prompt-using phases more consistent. Moreover, the ICL capability of LLMs allows them to make predictions based on a few labeled data as demonstrations. Theoretical studies have linked ICL behaviors to gradient descent algorithms, suggesting that ICL can be viewed as implicitly constructing a model that performs empirical risk minimization based on demonstrations, and then uses this model to predict unlabeled query data (Akyürek et al., 2022; Von Oswald et al., 2023). By simultaneously learning both the pseudo labels and the prompt during the prompt-learning phase, we ensure consistent prediction using the learned prompt and pseudo-labeled demonstrations during the prompt-using phase, ultimately generating reliable pseudo labels for the downstream task. We tested the proposed algorithm in benchmark applications and the results showed the effectiveness of the proposed method.

## 2 RELATED WORK

**Supervised Fine-tuning with Open-sourced Models.** Many parameter-efficient fine-tuning techniques were proposed to fine-tune an open-sourced transformer-based LLM with labeled data in the downstream task. Adapter-based fine-tuning methods introduce an adapter module into transformer, allowing for fine-tuning without changing the pre-trained parameters (Houlsby et al., 2019; He et al., 2021; Lei et al., 2023). Soft prompt fine-tuning algorithms append soft prompts or prefix vectors to the input embeddings or hidden states during fine-tuning (Li & Liang, 2021; Liu et al., 2023; Wang et al., 2022). Re-parameterized fine-tuning methods use low-rank transformation to reduce the number of trainable parameters while allowing operation with pre-trained parameters (Hu et al., 2021; Zhang et al., 2023; Yang et al., 2024). More recently, Chen et al. (2024) proposed a zero-order optimizer for large-scale deep neural networks. Malladi et al. (2023) proposed a memory-efficient zero-order optimizer to alleviate the heavy back-propagation in fine-tuning LLMs. However, all of these methods require that the open-source LLMs are first duplicated and then fine-tuned with labeled data in the downstream task, which is not applicable in our setting.

**Supervised Prompt Learning with Black-box Models.** Another line of work investigated prompt learning for the black-box LLMs released as commercial application programming interfaces. Sun et al. (2022) optimized the input prompt for adaptation using derivative-free optimization techniques such as the evolutionary algorithms. Diao et al. (2022) employed the policy gradient algorithms to optimize the input prompt to fine-tune the model. While these methods consider the black-box learning scenario of LLMs, they still require the labeled data to learn the prompt.

**Unsupervised Fine-tuning with Open-sourced Models.** Several recent studies on large vision-language models explored model fine-tuning using only leveraging unlabeled data in downstream tasks. These methods explore the parameters in the vision-language model to obtain class embeddings, which are then updated to align with the data distribution of the downstream task. For example, Huang et al. (2022) used the text encoder to generate reliable pseudo labels and then fine-tuned the model. Shu et al. (2022) augmented the unlabeled image and used the image encoder to generate reliable pseudo labels and then fine-tuned the model. Tanwisuth et al. (2023) and Ma et al. (2023) explored the class embeddings inside the LLMs to adapt the downstream tasks. Although these algorithms only require unlabeled data, they assume an open-source setting where model parameters are accessible during adaptation.

## 3 OUR APPROACH

In this section, we first formulate the unsupervised prompt learning problem. Next, we propose a new prompt learning objective with pseudo-labeled demonstrations that simultaneously learns the prompt and pseudo labels for the downstream task. Lastly, we introduce the optimization strategy and provide implementation details.

### 3.1 PROBLEM FORMULATION

In this part, we first formulate the learning problem and introduce the notations. Let $\mathbf{x}_l \in \mathcal{X}$ be the $l$-th query in the unlabeled data of size $n$, where $\mathcal{X}$ is the textual space. We denote by $\mathbf{z} = [z_1, \ldots, z_i, \ldots, z_m] = [V[j_1], \ldots, V[j_i], \ldots, V[j_m]]$ the discrete prompt of length $m$, where $V$ is the vocabulary set containing $N$ tokens, and $z_i = V[j_i]$ is the $i$-th token in $\mathbf{z}$, corresponding to the $j_i$-th token in $V$. Considering the in-context learning capability of LLMs, we define the black-box LLM by the function $f(\cdot, \cdot, \cdot) : (\mathbf{x}, \mathbf{z}, D) \mapsto \mathbf{y}$, where $\mathbf{x}$ is the unlabeled query, $\mathbf{y} \in \mathbb{R}^C$ is the logit vector of length $C$ over a set of label words, $\mathbf{z}$ is the prompt, and $D = \{(\mathbf{x}_k, \widehat{y}_k)\}_{k=1}^K$ is a set of in-context demonstrations of size $K$ selected from the downstream task. $D$ can be an empty set, e.g., $f(\cdot, \cdot, \emptyset)$, indicating that the LLM is used directly for prediction without in-context demonstrations.

For each discrete token $z_i = V[j_i]$, where $i = 1, \ldots, m$, we assume that it is sampled independently from the vocabulary set $V$ according to a categorical distribution. Specifically, we sample $z_i$ by first sampling the vocabulary index $j_i \sim \mathbf{p}_i$, where $\mathbf{p}_i = [p_{i,1}, \ldots, p_{i,N}]$ is a categorical distribution over the vocabulary set, with $\mathbf{p}_i \in \mathcal{C}$ and $\mathcal{C} = \{\mathbf{p} \in \mathbb{R}^N : \sum_{j=1}^N p_j = 1, p_j \geq 0 \text{ for } j = 1, \ldots, N\}$. Since each $\mathbf{z}_i$ is independently sampled from a categorical distribution for each $i$, where $j_i \sim \mathbf{p}_i$, the joint probability of the entire discrete prompt is given by $\Pi_{i=1}^m \Pr(z_i) = \Pi_{i=1}^m p_{i,j_i}$. Given a set

of unlabeled data for the downstream task and a black-box LLM $f$, our goal is to learn a prompt $\mathbf{z}$ that minimizes the prediction loss on the downstream tasks. When the in-context demonstrations are unavailable during inference, we can directly use $f(\cdot, \cdot, \emptyset)$ for prediction.

Since the downstream task is unlabeled, we propose to first identify reliable pseudo-labeled data to learn the prompt. Given the black-box access to LLMs, we rely on the output of the LLM, i.e., the logit vector over the label words, to estimate the confidence score of each prediction, selecting high-confidence predictions as reliable pseudo-labeled data. Specifically, we denote the linguistic confidence of the LLM's output for $\mathbf{x}_l$ by $\Pr[y|\mathbf{x}_l, \mathbf{z}; t] = [f^{(t)}(\mathbf{x}_l, \mathbf{z}, \emptyset)/\sum_{c=1}^{C}[f^{(t)}(\mathbf{x}_l, \mathbf{z}, \emptyset)]_c]_y, y = 1, \ldots, C$, where $[\cdot]_c$ refers to the $c$-th element in the vector $[\cdot]$ and $f^{(t)}(\cdot, \cdot, \cdot)$ represents the LLM with temperature $t$. Based on this, we can then define the confidence score for the prediction on $\mathbf{x}_l$, denoted as $c_l \in [0, 1]$. We first define the *average linguistic confidence* of a prediction, which is used to select reliable pseudo-labeled data. We define the average linguistic confidence score as:

$$c_l^{\text{LG}} = \max_{y \in \{1, \ldots, C\}} \left\{ \frac{1}{|T|} \sum_{t \in T} \Pr[y|\mathbf{x}_l, \mathbf{z}; t] \right\},$$

where $T$ is a set of the temperature candidates.

We also adopt strategies from previous seminal works to compute the confidence score. These approaches aim to reduce class-specific prior biases by using random text and averaging the estimates over multiple times (Zhao et al., 2021; Fei et al., 2023). Specifically, we adopt the approach from the work of Fei et al. (2023) to estimate the output confidence. Let $\mathbf{x}_{\text{rand}}$ be the random context sampled from the downstream task corpus. We then define the *bias-reduced confidence* score as:

$$c_l^{\text{RD}} = \max_{y \in \{1, \ldots, C\}} \left\{ \frac{\Pr[y|\mathbf{x}_l, \mathbf{z}; 0]}{\frac{1}{|T|} \sum_{t \in T} \Pr[y|\mathbf{x}_{\text{rand}}, \mathbf{z}; t]} \right\}.$$

It is important to note that the *confidence estimation mechanism used in our approach is flexible and not the primary focus of this work*. In this paper, we propose a direct confidence score estimation method by averaging the linguistic confidence, along with a state-of-the-art algorithm (Fei et al., 2023), to generate confidence scores and identify reliable pseudo-labeled data whose score is higher than a threshold. Confidence estimation for LLMs has received considerable attention in recent years. A comprehensive overview can be found in the survey by Geng et al. (2024).

### 3.2 UNSUPERVISED PROMPT LEARNING WITH PSEUDO-LABELED DEMONSTRATIONS

In this part, we propose the unsupervised prompt learning algorithm. To achieve accurate label predictions for the downstream task, we propose to learn the prompt and pseudo labels simultaneously, ensuring consistency between the prompt-learning and prompt-using phases. To predict unlabeled data in a downstream task, a direct approach is to assign pseudo labels to the unlabeled data, identify reliable pseudo-labeled data, learn a prompt based on these reliable pseudo labels, and then use the learned prompt for prediction with these reliable pseudo-labeled data. Specifically, we can learn the prompt by optimizing the following objective:

$$\arg\min_{\mathbf{z} \in \mathcal{Z}} \sum_{l=1}^{n} \mathbb{1}[c_l \geq \gamma] \cdot \ell(f(\mathbf{x}_l, \mathbf{z}, \emptyset), f(\mathbf{x}_l, \emptyset, \emptyset)),$$

where $\mathbb{1}[\cdot]$ is the indicator function, $\ell(\cdot, \cdot)$ is the loss function and $\gamma \in [0, 1]$ is a confidence threshold for selecting reliable pseudo-labeled data. In this paper, we consider two types of loss functions: hinge loss and cross entropy.

Although learning prompts with reliable pseudo-labeled data is feasible, the prompt-training and the prompt-using stages are not consistent, and the number of such reliable data may be limited, and relying solely on them has a risk of severe overfitting. To handle this problem, we propose to learn the pseudo labels for the entire unlabeled dataset and the prompt simultaneously, inspired by the ICL capabilities of LLMs (Brown et al., 2020; Wei et al., 2023). The ICL capabilities allow LLMs to implicitly learn a classifier from labeled demonstrations and apply it to predict other unlabeled data in downstream tasks. We illustrate the proposed algorithm in Figure 2. We first reliable pseudo-labeled data based on the output confidence. We then simultaneously learn the prompt and the pseudo

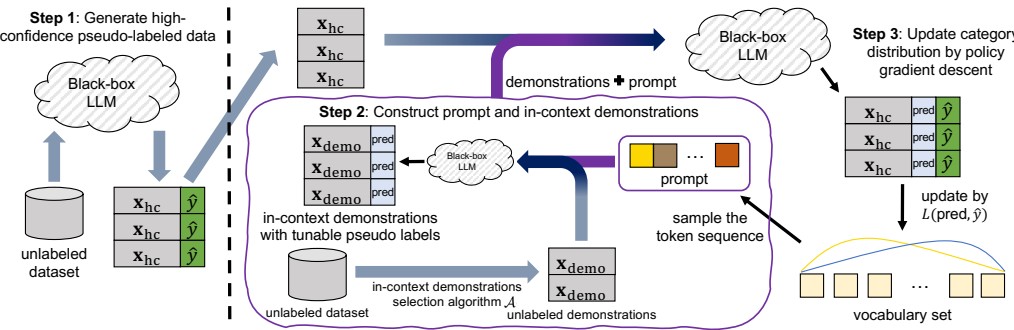

**Figure 2:** An illustration of proposed algorithm. We begin by identifying reliable pseudo-labeled data based on the output of the LLM, selecting the high-confidence predictions as reliable data. We then construct demonstrations for each sample, consisting of a set of data from the downstream task, selected by a specific algorithm and assigned pseudo labels predicted by the LLM, guided by the prompt and their corresponding demonstrations. After that, we align the predictions based on the demonstrations and prompt with the original pseudo labels on the reliable data. The prompt is a sequence of tokens sampled from corresponding categorical distributions over a vocabulary set and updated by the policy gradient descent algorithm.

labels for the unlabeled data by ensuring that, for the reliable pseudo-labeled data, when the LLM is presented with the prompt and other pseudo-labeled data as in-context demonstrations, its predictions remain consistent with the original pseudo labels.

In the following, we introduce the main loss in our learning objective, which is defined as follows:

$$L_{\mathrm{main}}(\mathbf{z}) = \sum_{l=1}^{n} \mathbb{1}[c_l \geq \gamma] \cdot \ell\left(f(\mathbf{x}_l, \mathbf{z}, D_l), f(\mathbf{x}_l, \emptyset, \emptyset)\right), \tag{1}$$

where $D_l$ is a set of in-context demonstrations for the query $\mathbf{x}_l$ selected by algorithm $\mathcal{A}(\mathbf{x}_l; \mathbf{z})$, with the pseudo labels of these demonstrations determined by the prompt $\mathbf{z}$. Specifically, $\mathcal{A}(\mathbf{x}_l, \mathbf{z})$ outputs a set of pseudo-labeled demonstrations selected from the downstream task:

$$D_l = \left\{ (\mathbf{x}_k, \arg\max_c [f(\mathbf{x}_k, \mathbf{z}, \mathcal{D}_k)]_c) | \mathbf{x}_k \in S_l \right\}_{k=1}^{K},$$

where $[\cdot]_c$ is the $c$-th element in the vector $[\cdot]$, indicating the $c$-th class in the label space. Many algorithms have been proposed for selecting demonstrations. Following the seminal works on in-context demonstrations selection (Liu et al., 2022; Min et al., 2022), we select the $K$-nearest samples as in-context demonstrations for each sample $\mathbf{x}_l$, denoted as $S_l$. We formulate $S_l$ as follows:

$$S_l = \underset{\{k_j\}_{j=1}^{K} \subset \{1, \ldots, n\}}{\arg\min} \sum_{j=1}^{K} d(\mathbf{x}_l, \mathbf{x}_{k_j}), \tag{2}$$

where $d(\cdot, \cdot)$ is a distance measure between two data. We follow the same procedure outlined in the work of Liu et al. (2022), introducing a sentence encoder $\theta(\cdot)$ and defining the distance as $d(\mathbf{x}_l, \mathbf{x}_k) = \|\theta(\mathbf{x}_l) - \theta(\mathbf{x}_k)\|_2$.

Since we generate the prompt by sampling each discrete token according to a learned categorical distribution during testing, our goal is to ensure a stable prompt during inference. Specifically, we measure the Shannon entropy (Shannon, 2001) of the categorical distribution for each $i$-th token as follows:

$$H(\mathbf{p}_i) = -\sum_{j=1}^{N} p_{i,j} \log p_{i,j},$$

where $\mathbf{p}_i$ is the probability distribution. Consequently, we define the loss function for the token probability $\mathbf{p}$ as follows:

$$L(\mathbf{p}) = \mathbb{E}_{\mathbf{z} \sim \mathbf{p}} \left[ L_{\mathrm{main}}(\mathbf{z}) \right] + \alpha \sum_{i=1}^{m} H(\mathbf{p}_i), \tag{3}$$

where $\alpha \geq 0$ is a hyperparameter.

---

**Algorithm 1** Prompt learning with Pseudo-labeled Demonstrations (PPD)

---

1: Set vocabulary set $V$, total number of iterations $T$ and total number of sampling $I$
2: Initialize categorical probability distribution $\{\mathbf{p}_i\}_{i=1}^m$
3: **for** $t = 1$ **to** $T$ **do**
4:     Sample a mini-batch data from the unlabeled dataset
5:     **for** $k \leq I$ **do**
6:         Sample $j_i^{(k)} \sim \mathrm{Cat}(\mathbf{p}_1^t), \ldots, j_n^{(k)} \sim \mathrm{Cat}(\mathbf{p}_m^t)$
7:         $\mathbf{z}^{(k)} = [z_1^{(k)}, \ldots, z_m^{(k)}] = [V[j_1^{(k)}], \ldots, V[j_m^{(k)}]]$
8:     **end for**
9:     **for** $i \leq m$ **do**
10:        Update $\mathbf{p}_i$ according to Equation (4)
11:     **end for**
12: **end for**

---

### 3.3 IMPLEMENTATION

Here, we introduce the optimization algorithm for unsupervised prompt learning. Given the black-box setting of the LLMs in our problem, we cannot access training gradients or use back-propagation to learn the prompt. To handle the challenge of optimizing the discrete prompt, we employ the *variance-reduced policy gradient estimator* (VR-PGE) (Williams, 1992; Zhou et al., 2021; Diao et al., 2022), a well-developed policy gradient algorithm in discrete optimization.

In particular, the VR-PGE algorithm first estimates the gradient for each categorical distribution and then optimizes it by forward propagation. Specifically, the gradient of the $i$-th discrete token is

$$
\nabla_{\mathbf{p}_i}[L(\mathbf{p})] = \nabla_{\mathbf{p}_i} \left[ \mathbb{E}_{\mathbf{z} \sim \mathbf{p}}[L_{\mathrm{main}}(\mathbf{z})] + \alpha \sum_{i=1}^m H(\mathbf{p}_i) \right]
$$

$$
= \int \Pr(\mathbf{z}) L_{\mathrm{main}}(\mathbf{z}) \nabla_{\mathbf{p}_i} \log \Pr(\mathbf{z}) \mathrm{d}\mathbf{z} + \alpha \nabla_{\mathbf{p}_i} H(\mathbf{p}_i)
$$

$$
= \mathbb{E}_{\mathbf{z}}[L_{\mathrm{main}}(\mathbf{z}) \nabla_{\mathbf{p}_i} \log \Pi_{j=1}^m \Pr(z_j)] + \alpha \nabla_{\mathbf{p}_i} H(\mathbf{p}_i)
$$

$$
= \mathbb{E}_{\mathbf{z}}[L_{\mathrm{main}}(\mathbf{z}) \nabla_{\mathbf{p}_i} \log \Pr(z_i)] + \alpha \nabla_{\mathbf{p}_i} H(\mathbf{p}_i).
$$

The $j$-th component of $\nabla_{\mathbf{p}_i} \log \Pr(z_i)$ is given by $\nabla_{p_{i,j}} \log \Pr(z_i) = 1/p_{i,j_i}$ when $j = j_i$, and $\nabla_{p_{i,j}} \log \Pr(z_i) = -1/p_{i,j_i}$ when $j \neq j_i$. The $j$-th component of $\nabla_{\mathbf{p}_i} H(\mathbf{p}_i)$ is given by $\nabla_{p_{i,j}} H(\mathbf{p}_i) = -\log p_{i,j} - 1$. The estimated gradient is calculated using the VR-PGE, that is,

$$
\nabla_{\mathbf{p}_i}^{\mathrm{vr}}[L(\mathbf{p})] = \frac{1}{I-1} \sum_{k=1}^I \left( L_{\mathrm{main}}(\mathbf{z}^{(k)}) - \frac{1}{I} \sum_{k'=1}^I L_{\mathrm{main}}(\mathbf{z}^{(k')}) \right) \nabla_{\mathbf{p}_i} \log \Pr(z_i) + \alpha \nabla_{\mathbf{p}_i} H(\mathbf{p}_i),
$$

where $\mathbf{z}^{(k)}, k = 1, \ldots, I$ and $\mathbf{z}^{(k')}, k' = 1, \ldots, I$ are sampled independently from $\mathbf{p}$ and $I$ is the sample size.

Then we update the categorical distribution by a projected stochastic gradient descent algorithm:

$$
\mathbf{p}_i \leftarrow \mathrm{proj}_{\mathcal{C}}(\mathbf{p}_i - \eta \cdot \nabla_{\mathbf{p}_i}^{\mathrm{vr}}[L(\mathbf{p})]), i = 1, \ldots, m, \tag{4}
$$

where $\eta > 0$ is the learning rate and $\mathrm{proj}_{\mathcal{C}}$ is the projection function onto $\mathcal{C}$.

We summarize the learning procedure in Algorithm 1. We first initialize the categorical distribution for each token in the prompt. In each round, we sample a batch of unlabeled data and perform $I$ iterations of sampling to compute the VR-PGE. Specifically, at the $k$-th iteration, we first sample the sequence of tokens $\mathbf{z}_i, i = 1, \ldots, m$, according to the distribution $\mathbf{p}_i, i = 1, \ldots, m$ to form the prompt $\mathbf{z}$, and combine it with the unlabeled query and demonstrations. Then we present $[\mathbf{x}_l, \mathbf{z}, D]$ to the black-box LLM $f(\cdot, \cdot, \cdot)$ and obtain the prediction. We then compute the loss according to Equation (3) and update the categorical distributions by Equation (4). After learning the prompt and pseudo labels for the downstream task, we can directly use the pseudo-labeled data for further fine-tuning of the model by using the commercial fine-tuning service. We can use the learned categorical distributions to formulate the prompt to predict new unlabeled data.

We construct the vocabulary $V$ following previous work (Shin et al., 2020; Diao et al., 2022). We adopt *point-wise mutual information* (PMI) to construct the vocabulary of candidate prompt tokens in an unsupervised manner. For each sentence in the downstream task, we compute the PMI by $\text{PMI}(x_1, x_2) = \log(\Pr(x_1, x_2)/(\Pr(x_1)\Pr(x_2)))$, where $x_1$ and $x_2$ are two adjacent words in the sentence, and $\Pr(x)$ is the probability of an n-gram $x$. Therefore, the sentence is segmented based on the PMI scores. We obtain a list of n-grams $V$ by extracting those consecutive words after segmentation and with a frequency of at least $\delta$. As for $N$, we choose $N$ between 50 and 200.

## 4 EXPERIMENTS

In this section, we evaluate the proposed algorithm with contenders using various benchmark datasets. Next, we conduct ablation studies to verify the effectiveness of each component in our approach.

### 4.1 EXPERIMENTAL SETUP

**Datasets.** We use GLUE datasets (Wang et al., 2018): CoLA, SST-2, QQP, MRPC, MNLI, WNLI, RTE; and MMLU datasets (Hendrycks et al., 2021): astronomy (AST), high-school-computer-science (HSCS), high-school-mathematics (HSM), college-mathematics (Cmath), college-computer-science (CCS), college-medicine (CMed), management (MAN), marketing (MAR), all-random (RND).

**Contenders.** Since we are the first to investigate the problem of unsupervised prompt learning, there are no other methods to be directly used to compare with our approach. We compare the proposed algorithm with four contenders. The first one is *Direct*, that is, we directly use the LLM to generate the prediction. We also take two black-box LLM prompt learning algorithms into comparison. Since we only have unlabeled data in the downstream task, we first select reliable pseudo-labeled data and then use the black-box LLM prompt learning algorithms to learn the prompt. Specifically, *BDPL* (Diao et al., 2022) uses a policy gradient descent method to optimize the categorical distribution of each token in the prompt. *RLprompt* (Deng et al., 2022) formulates a parameter-efficient policy network that generates the optimized discrete prompt after training with reward. We denote by "RD" that the algorithm uses bias-reduced confidence to generate the reliable pseudo-labeled data, while "LG" denotes that we use average linguistic confidence to generate the reliable data. We also introduce the contender *ICL* (Liu et al., 2022) that use the selected high-confidence pseudo-labeled data as in-context demonstrations to predict the remaining unlabeled data.

**Implementation Details.** In all experiments, we used GPT-4[2] and gpt-4o-mini[3], provided by OpenAI, where gpt-4o-mini is much cheaper than GPT-4. During preprocessing, we used the API to query the black-box LLM with manual templates as specified in Appendix A.1. Confidence scores were calculated based on the log probabilities returned during the query. For BDPL, we applied the original BDPL algorithm (Diao et al., 2022), using pseudo labels as actual labels and setting high hyperparameters, as specified in Table 4 in Appendix A.2. The same approach was used for the RLPrompt algorithm and the ICL algorithm.

### 4.2 PERFORMANCE COMPARISON ON BENCHMARKS

In this part, we compare the proposed algorithm with other contenders on benchmark datasets. To ensure a fair comparison with supervised prompt-tuning methods, we use the Cross Entropy loss as the training loss for the BDPL, RLPrompt and PPD algorithms, as it demonstrates relatively better performance compared to the Hinge loss. For the BDPL and RLPrompt algorithms, we use the default parameter settings from their original papers. For the ICL algorithm, we apply the same in-context selection method and the number of demonstrations used in our proposed PPD algorithm. For the PPD algorithm, we set the number of in-context demonstrations to 5 and $\alpha$ to $2e^{-5}$. The confidence threshold is set to $\gamma = 0.7$ in the proposed PPD algorithm, and we report the results using the optimal confidence threshold for the contenders BDPL, RLPrompt, and ICL algorithms.

We report the comparison results of the proposed PPD algorithm with other contenders on benchmark datasets in Table 1 and Table 2. Our proposed PPD algorithm outperforms almost all other contenders

---

[2]https://platform.openai.com/docs/models/gpt-4-turbo-and-gpt-4
[3]https://platform.openai.com/docs/models/gpt-4o-mini

**Table 1:** Performance comparisons of the proposed PPD algorithm with other contenders on the GLUE dataset. For each dataset, 5 tests are conducted, and the average accuracy (%) as well as the standard deviation are presented. The best result of each dataset is emphasized in bold.

| Method | MNLI | QQP | SST-2 | MRPC | CoLA | WNLI | RTE |
|---|---|---|---|---|---|---|---|
| Direct | 91.6 ± 2.4 | 71.3 ± 1.1 | 89.5 ± 1.6 | 90.8 ± 2.1 | 69.5 ± 1.8 | 90.7 ± 1.7 | 92.8 ± 1.3 |
| ICL (LG) | 89.3 ± 1.7 | 67.5 ± 2.3 | 87.9 ± 0.6 | 88.7 ± 2.3 | 67.6 ± 1.3 | 88.7 ± 0.7 | 89.2 ± 0.8 |
| ICL (RD) | 90.2 ± 2.1 | 68.3 ± 2.1 | 88.3 ± 0.8 | 89.9 ± 1.6 | 65.5 ± 2.4 | 87.2 ± 1.8 | 88.5 ± 1.1 |
| BDPL (LG) | 92.3 ± 1.7 | 71.7 ± 1.9 | 91.2 ± 1.8 | 91.7 ± 1.2 | 69.8 ± 2.3 | 90.4 ± 1.3 | **93.1 ± 1.0** |
| BDPL (RD) | **92.5 ± 1.8** | 72.3 ± 1.7 | 90.2 ± 2.3 | 92.1 ± 0.9 | 67.2 ± 3.1 | 88.7 ± 2.1 | 91.8 ± 1.4 |
| RLprompt (LG) | 91.9 ± 2.1 | 69.7 ± 2.9 | 88.6 ± 2.5 | 88.9 ± 2.5 | 70.3 ± 2.2 | 90.6 ± 1.2 | 90.1 ± 1.5 |
| RLprompt (RD) | **92.5 ± 0.6** | 69.5 ± 3.2 | 89.5 ± 2.1 | 89.8 ± 1.9 | 69.7 ± 1.4 | 88.4 ± 1.9 | 89.9 ± 1.7 |
| PPD (LG) | 91.8 ± 2.3 | 71.5 ± 2.3 | **92.1 ± 2.4** | **90.9 ± 1.6** | **70.3 ± 1.7** | **91.3 ± 1.9** | 92.8 ± 2.1 |
| PPD (RD) | 92.1 ± 1.9 | **72.3 ± 2.1** | 91.5 ± 1.2 | 89.9 ± 1.8 | 69.0 ± 1.2 | 90.7 ± 1.5 | 92.1 ± 1.7 |

**Table 2:** Performance comparisons of the proposed PPD algorithm with other contenders on the MMLU dataset. For each dataset, 5 evolutions are conducted, and the average accuracy (%) as well as the standard deviation are presented. The best result of each dataset is emphasized in bold.

| Method | MAR | MAN | HSM | HCS | CMed | CMath | CCS | AST | RND |
|---|---|---|---|---|---|---|---|---|---|
| Direct | 90.4 ± 2.1 | 76.6 ± 1.5 | 50.7 ± 3.1 | 90.6 ± 2.9 | 61.7 ± 1.8 | 40.5 ± 4.3 | 68.2 ± 2.5 | **89.5 ± 2.7** | 68.6 ± 1.2 |
| ICL (LG) | 87.9 ± 2.3 | 75.1 ± 1.2 | 45.9 ± 3.8 | 87.5 ± 1.5 | 58.6 ± 4.3 | 37.4 ± 3.3 | 68.3 ± 2.5 | 88.9 ± 1.7 | 67.9 ± 1.5 |
| ICL (RD) | 88.5 ± 1.8 | 76.3 ± 0.9 | 47.2 ± 2.4 | 88.9 ± 2.2 | 58.2 ± 3.5 | 39.9 ± 2.6 | 69.3 ± 1.4 | 86.3 ± 2.4 | 68.1 ± 1.3 |
| BDPL (LG) | 90.1 ± 1.6 | 78.3 ± 1.5 | 51.6 ± 2.8 | 90.4 ± 3.0 | **62.1 ± 2.2** | 42.7 ± 3.3 | 70.1 ± 1.3 | 88.7 ± 2.2 | 70.1 ± 2.4 |
| BDPL (RD) | 90.7 ± 2.0 | 79.2 ± 1.1 | 53.1 ± 1.9 | 92.3 ± 2.2 | 61.5 ± 1.8 | 44.1 ± 2.9 | 71.5 ± 1.9 | 85.3 ± 3.8 | 70.5 ± 1.9 |
| RLprompt (LG) | 90.3 ± 2.5 | 77.3 ± 1.8 | 51.2 ± 2.5 | 89.6 ± 2.4 | 60.7 ± 2.3 | 40.7 ± 2.9 | 69.5 ± 2.4 | 89.3 ± 2.2 | 70.3 ± 1.9 |
| RLprompt (RD) | 91.1 ± 0.7 | 78.5 ± 0.8 | 54.3 ± 1.2 | 91.1 ± 2.3 | 59.5 ± 3.1 | 43.0 ± 1.4 | 70.9 ± 1.7 | 87.7 ± 2.9 | 70.6 ± 1.5 |
| PPD (LG) | 91.5 ± 1.3 | 78.4 ± 2.1 | 52.6 ± 2.4 | 91.6 ± 2.8 | 61.9 ± 2.9 | 43.5 ± 2.6 | 70.6 ± 2.1 | 88.8 ± 2.6 | 71.9 ± 2.5 |
| PPD (RD) | **91.9 ± 0.6** | **79.8 ± 1.5** | **53.5 ± 1.7** | **92.9 ± 1.5** | 61.4 ± 2.5 | **44.9 ± 1.7** | **72.8 ± 1.1** | 86.1 ± 1.6 | **72.5 ± 2.1** |

across the benchmark datasets. Compared with the Direct algorithm, our approach achieved superior performance on nearly all datasets, demonstrating our success in leveraging unlabeled data to learn an effective prompt for the downstream task and obtain more accurate predictions. The ICL algorithm even underperforms the Direct algorithm, indicating the need to explore other unlabeled data in the learning phase. Furthermore, our algorithm outperforms the BDPL and RLPrompt algorithms, highlighting the importance of introducing in-context demonstrations during the learning phase, as we further utilize pseudo-labeled data in the downstream tasks for prompt learning.

### 4.3 IN-CONTEXT DEMONSTRATIONS SELECTION

In this part, we evaluate the impact of varying the number of in-context demonstrations and the training loss used in our proposed algorithm. For in-context demonstrations selection, we select the $K$-nearest samples for each instance, following the distance measurement used by Liu et al. (2022).

We learn the prompt using different numbers of in-context demonstrations and loss functions, and present the comparison results in Table 3. It can be observed that all versions of the PPD algorithm, regardless of the number of in-context demonstrations, outperform the Direct algorithm and the ICL algorithm with different $K$, highlighting the benefit of leveraging pseudo-labeled data for unsupervised prompt learning in the downstream task. Among all tested values of $K$, setting $K$ to 5 and using the Cross Entropy loss yields the most satisfactory performance.

### 4.4 ABLATION STUDIES

In this part, we analyze several aspects of the proposed PPD algorithm. Specifically, we investigate the effect of different mechanisms for generating reliable pseudo-labeled data, the effect of the entropy minimization term in the optimization process, and the influence of the hyperparameters in the proposed PPD algorithm.

**Reliable Pseudo-labeled Data Selection.** We first investigate different mechanisms for generating reliable pseudo-labeled data and the effect of different confidence thresholds $\gamma$. We perform the experiments on the two datasets GLUE and MMLU with the average accuracy measurement and report the experimental results in Figure 3(a). We can observe that using RD and setting the confidence threshold to $0.7$ gives the best performance across all experiments. A small confidence threshold will introduce incorrect pseudo-labels and thus affect the performance. A relatively high confidence threshold will limit the size of the selected pseudo-labeled data and also cause performance

**Table 3:** Performance comparisons for different number of in-context demonstrations used unsupervised prompt learning on benchmark datasets. For each dataset, 5 evolutions are conducted, and the average accuracy (%) as well as the standard deviation are presented. The best result of each dataset is emphasized in bold.

| Method | MNLI | QQP | SST-2 | MRPC | CoLA | WNLI | RTE | RND |
|---|---|---|---|---|---|---|---|---|
| Direct | $91.6 \pm 2.4$ | $71.3 \pm 1.1$ | $89.5 \pm 1.6$ | $\mathbf{90.8 \pm 2.1}$ | $69.5 \pm 1.8$ | $90.7 \pm 1.7$ | $\mathbf{92.8 \pm 1.3}$ | $68.6 \pm 1.2$ |
| ICL ($k = 3$) | $89.3 \pm 1.9$ | $68.5 \pm 2.1$ | $88.9 \pm 2.4$ | $88.3 \pm 1.7$ | $66.4 \pm 2.3$ | $87.5 \pm 1.7$ | $88.3 \pm 1.2$ | $67.5 \pm 1.5$ |
| ICL ($k = 5$) | $90.5 \pm 0.8$ | $67.3 \pm 1.8$ | $88.4 \pm 1.8$ | $89.2 \pm 2.1$ | $67.2 \pm 1.7$ | $88.4 \pm 1.1$ | $88.9 \pm 0.9$ | $68.1 \pm 1.7$ |
| PPD ($k = 3$) | $91.5 \pm 2.1$ | $71.5 \pm 2.1$ | $90.3 \pm 1.7$ | $89.3 \pm 1.8$ | $68.8 \pm 1.5$ | $90.5 \pm 1.7$ | $92.1 \pm 2.0$ | $71.5 \pm 2.6$ |
| PPD ($k = 5$) | $\mathbf{92.3 \pm 1.7}$ | $\mathbf{72.1 \pm 1.9}$ | $\mathbf{92.1 \pm 2.2}$ | $90.2 \pm 1.9$ | $\mathbf{70.5 \pm 1.1}$ | $\mathbf{91.2 \pm 1.3}$ | $92.6 \pm 1.8$ | $\mathbf{72.7 \pm 2.2}$ |

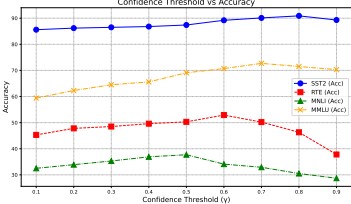

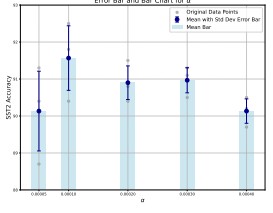

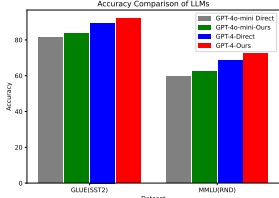

(a) Performance with different reliable data generation mechanisms and confidence threshold $\gamma$.

(b) Accuracy and standard deviation with different entropy minimization weights.

(c) Accuracy comparison with two black-box LLMs: GPT-4 and GPT-4o-mini.

**Figure 3:** Ablation studies for several aspects of the proposed PPD algorithm.

degradation. Therefore, it is suggested to use the bias-reduced confidence to generate the initial reliable pseudo-labeled data and set the confidence threshold to 0.7 in practice.

**Effect of Entropy Minimization.** We then examine the effect of the entropy minimization term with various $\alpha$ during inference on the SST-2 dataset. Since each discrete token is sampled from a corresponding categorical distribution to form the prompt, the proposed entropy minimization term aims to generate a more stable prompt during inference.

We report the experimental results in Figure 3(b). We can observe that, with the increasing of $\alpha$, the variance is reducing. This is because if we do not use the entropy minimization term during optimization, the learned category distributions can even be flatten, making a random generation of certain tokens in the prompts during inference time. Therefore, we introduce the entropy minimization term for a stable prompt generation during inference and thus the variance is reduced.

**Different black-box LLMs.** Finally, we evaluate the performance of the proposed algorithm with different black-box LLMs. Experiments are conducted on three datasets: GLUE, MMLU, using average accuracy as the metric. We report the results for GPT-4 and GPT-4o-mini in Figure 3(c). For a fair comparison, we set the number of demonstrations to 5 and use the Cross Entropy loss.

We compare these results with those of the Direct algorithm. The proposed PPD algorithm outperforms the Direct algorithm with both GPT-4 and GPT-4o-mini, demonstrating the general applicability of our approach across different popular LLMs.

## 5 CONCLUSION

In this paper, we investigate unsupervised prompt learning for classification with black-box LLMs, where we learn the prompt itself and the pseudo labels of unlabeled data simultaneously. After first identifying initial reliable pseudo-labeled data using the LLM, we propose a novel learning objective where we assign pseudo labels to other unlabeled data based on the prompt, allowing the pseudo-labeled data to serve as in-context demonstrations alongside the prompt. These in-context demonstrations are important: they are previously involved when the prompt is used for prediction, while they are not involved when the prompt is trained; thus, taking them into account during training makes the prompt-learning and prompt-using phases more consistent. Experiments on benchmark datasets demonstrate the effectiveness of our proposed algorithm. After unsupervised prompt learning and obtaining learned pseudo labels for the downstream task, we can then use the pseudo-labeled dataset for further fine-tuning by the owners of the black-box LLMs.

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

# A APPENDIX

## A.1 MANUAL TEMPLATES

The manual templates for each dataset are listed as follows. These manual templates follow the prompt and demonstrations.

**SST-2**:

    Review: { sentence }, Options: { options }. Answer:

**COLA**:

    Sentence: { sentence } Options: { options }. Answer:

**MNLI**:

    Premise: { premise }\nHypothesis: { hypothesis }\nOptions: {
        options }. Answer:

**QQP**:

    Question 1: { question1 }\nQuestion 2: { question2 }\nOptions: {
        options }. Answer:

**MRPC**:

    Sentence 1: { sentence1 }\nSentence 2: { sentence2 }\nOptions: {
        options }. Answer:

**RTE**:

    Premise: { sentence1 }\nHypothesis: { sentence2 }\nOptions: {
        options }. Answer:

**WNLI**:

    Sentence 1: { sentence1 }\nSentence 2: { sentence2 }\nOptions: {
        options }. Answer:

**CAIS/MMLU**:

    Question: { question }, Options: { options }. Answer:

## A.2 HYPERPARAMETERS SETTING

**Table 4:** Hyper Parameters for Each Dataset

| Dataset | MNLI | QQP | SST-2 | MRPC | CoLA | WNLI | RTE | MMLU |
|---|---|---|---|---|---|---|---|---|
| **Prompt Length** | 20 | 50 | 50 | 40 | 50 | 50 | 50 | 50 |
| **Learning Rate** | 2e-4 | 1e-4 | 4e-4 | 1e-4 | 2e-4 | 1e-4 | 1e-4 | 2e-5 |
| **# of Demonstrations** | 4 | 3 | 3 | 4 | 5 | 5 | 5 | 5 |

**Table 5:** Datasets with corresponding prompt lengths and learning rates.

