# OpenReview forum: "On Unsupervised Prompt Learning for Classification with Black-box Language Models"
_ICLR.cc/2025/Conference — ICLR 2025 Conference Withdrawn Submission_

### Official Review · Reviewer_szmN · 2024-10-29

**Soundness:** 2
**Presentation:** 2
**Contribution:** 1
**Rating:** 3
**Confidence:** 3

**Summary:**

This paper proposes an unsupervised prompt learning method for text classification with black-box LLMs, combining unlabeled data, in-context learning (ICL), and prompt tuning to enhance classification performance. While the work is well-presented in detail, its contribution is relatively incremental, with limited novelty.

**Strengths:**

1. The paper is well-organized and clearly presented.

2. Extensive datasets from GLUE and MMLU are used to validate the proposed PPD approach.

**Weaknesses:**

1. Limited Novelty (major concern): The approach lacks novelty, as key components—selecting confident pseudo-labeled data, initializing prompts with categorical distribution, using KNN for demonstrations, and updating parameters with cross-entropy and VR-PGE—are directly adopted from existing methods with minimal modifications. The contribution appears to be a straightforward combination of established techniques, without addressing specific challenges in these components or their integration.

2. Inaccurate Experimental Analysis: The claim in Lines 414–416 that PPD (k=3) consistently outperforms Direct (Table 3) is inaccurate; Direct performs better on MNLI, MRPC, CoLA, WNLI, and RTE.

3. Incomplete Ablation Study: The ablation study lacks thoroughness. Beyond analyzing hyperparameters, loss functions, and LLMs, the impact of omitting individual PPD components should be assessed. Although some data is in Tables 1 and 2, further explicit analysis is needed.

4. Dataset Statistics (minor concern): Dataset statistics should be included.

5. Font Size in Figures (minor concern): The font size in all images, particularly in Figure 3 (a, b, and c), should be increased for readability.

**Questions:**

Please see the weakness.

**Details Of Ethics Concerns:**

N.A.

---

### Official Review · Reviewer_tCZm · 2024-11-02

**Soundness:** 2
**Presentation:** 1
**Contribution:** 1
**Rating:** 1
**Confidence:** 5

**Summary:**

This paper proposes to perform unsupervised prompt learning for classification with black-box LLMs, where the learning parameters include both the prompt and the pseudo labels of unlabeled data. (1) the prompt is modeled as a sequence of discrete tokens, each token having its to-be-learned categorical distribution. (2) To learn pseudo labels, authors first identify several reliable pseudo-labeled data and then use these data as demonstrations for ICL to annotate more unlabeled data. By performing prompt training using these data, the model can perform downstream tasks. Experiments on various benchmark datasets show the effectiveness of the proposed method.

**Strengths:**

This paper has several notable strengths, which are listed below:

1. The overall motivation behind this paper is sound. The LLMs have demonstrated superior annotation capabilities compared to humans, making it a logical step to consider using LLMs for labeling a larger portion of unlabeled data.

2. The paper features effective visual illustrations. The figures clearly convey the core concepts, enhancing the understanding.

3. The experiments conducted in this paper are based on the current state-of-the-art LLMs and incorporate a multi-perspective analysis, which appears to be thorough and comprehensive.

**Weaknesses:**

Several weaknesses need to be addressed.

(1) When referring to "black-box LLMs" I believe the authors mean "in-house LLMs" in contrast to open-source LLMs. It would be helpful if they could clarify this key distinction.

(2) It is a fact that given unlabeled data, one can perform various tasks. For example, with the text "I like the movie. I do enjoy the storyline," one could perform sentiment analysis by labeling it as "positive," and one could also conduct natural language inference by labeling it as "entailment." The paper does not mention an initial human labeling process to indicate specific tasks, as it directly starts with several reliable pseudo-labeled data generated by LLMs. I'm curious if this implies that the biases of the LLMs influenced the defined tasks.

(3) It would be beneficial if the authors could further explain the prompt training process, detailing how to train both the prompts and pseudo-labels simultaneously.

(4) This idea seems somewhat similar to works related to self-learning (e.g., iPET, iterative prompt tuning). The key difference appears to be that the data used in this approach is annotated by LLMs. It would help if the authors could clarify the key differences between their work and these other approaches.

**Questions:**

(1) One important question is why we need "unsupervised prompt learning." Could prompting with less but higher-quality data serve as an alternative solution? Additionally, might regular fine-tuning with few-shot data be another viable option? If the authors intend to highlight the advantages of this method, it would be beneficial to include relevant experiments and comparative analysis.

---

### Official Review · Reviewer_4Nqa · 2024-11-04

**Soundness:** 3
**Presentation:** 3
**Contribution:** 2
**Rating:** 5
**Confidence:** 3

**Summary:**

This paper introduces an unsupervised prompt learning method tailored for classification tasks with black-box large language models. It proposes a technique where the prompt and pseudo-labels are learned concurrently, leveraging pseudo-labeled data as in-context demonstrations. This approach contrasts with traditional methods that rely on labeled data for prompt learning. The authors first select high-confidence pseudo-labeled data using the LLM’s predictions and then use these for further prompt and label refinement, aiming to reduce inconsistencies between prompt-learning and prompt-using phases. The proposed method, termed Pseudo-labeled Prompt Demonstration (PPD), is evaluated on GLUE and MMLU benchmarks, showing improved performance over several baseline methods.

**Strengths:**

- The paper presents a unique approach to prompt learning without labeled data, which is particularly valuable in scenarios with limited labeled resources.
- The use of pseudo-labeled data as in-context demonstrations during training is a clever adaptation of LLM capabilities, making prompt training more consistent with usage.
- The paper conducts comprehensive evaluations on diverse datasets and includes several baseline comparisons, demonstrating the method’s performance across a wide range of tasks. GLUE is a bit outdated though.
- Detailed ablation studies illustrate the impact of various components, such as the choice of in-context demonstrations and confidence threshold for pseudo-labeling.

**Weaknesses:**

- The method would be more appreciated in a previous-year conference. However, as the paradigm has shifted, the proposed method is not technically novel or significant.
- The method’s performance depends on the accuracy of pseudo-labeling, which may be unreliable for challenging datasets or tasks with highly ambiguous labels. There’s a potential risk of propagating incorrect labels.
- As each training sample relies on in-context demonstrations, the approach may struggle with large datasets or scenarios requiring extensive pseudo-label generation, possibly leading to increased computational cost.
- Comparing this method with other non-prompt-based unsupervised classification techniques could provide additional insights.
- The optimization process, including sampling tokens and updating distributions with VR-PGE, could be difficult to reproduce or adapt to other LLM settings.

**Questions:**

- Can you elaborate on the computational costs involved, particularly with in-context demonstrations, and how they scale with larger datasets?

---

### Official Review · Reviewer_jK3D · 2024-11-04

**Soundness:** 2
**Presentation:** 2
**Contribution:** 2
**Rating:** 3
**Confidence:** 4

**Summary:**

This paper introduces an unsupervised prompt learning method using black-box, proprietary LLMs (without access to the model parameters). The method integrates several techniques: (1) pseudo-labeling of unlabeled data; (2) learning discrete prompt tokens; and (3) in-context prediction. Specifically, it first uses a black-box LLM to obtain pseudo-labeled data, then combines pseudo-labeled in-context examples and discrete prompt tokens sampled from a vocabulary to make predictions, and optimizes (1) an entropy term and (2) a consistency loss between the in-context predictions and the zero-shot predictions to update the policy of sampling prompt tokens. The main contribution of this paper lies in the proposed method that leverages proprietary LLMs (i.e., GPT-4 and GPT-4o-mini) to obtain high-quality pseudo-labeled data for downstream tasks.

**Strengths:**

Originality: This paper combines several existing ideas together to study a new setting of unsupervised prompt learning on unlabeled data. Although most components (i.e., discrete prompt learning, in-context prediction, pseudo-labeling) used in the proposed method are not new, integrating them is new.

Quality: The overall quality is decent. Experiments and analysis have been conducted on two popular benchmarks to evaluate the effectiveness of the proposed method.

**Weaknesses:**

Clarity: The clarity needs improvement. (1) The intuition of optimizing the consistency loss term and the entropy term needs to be clearly explained. See the detailed questions below. (2) The math notation can be further simplified to improve the readability.

Soundness: (1) The performance of the GPT-4 family on the two selected benchmarks is highly saturated, making it hard to see a significant improvement in the method over direct prompting. Direct prompting with GPT-4 on many tasks (like MNLI, SST-2, MRPC, WNLI, and RTE) even has ~90 percent accuracy.  It’d be better to use another benchmark dataset that has a relatively lower performance with direct prompting. (2) In Tables 2 & 3, ICL is consistently worse than Direct, which seems contradictory to the prior studies and this paper’s claim that in-context examples are helpful to PPD. (3) There are missing ablation studies, comparing PPD, PPD without the learned prompt tokens, and PPD without the in-context examples. This seems more important than the reported ablation studies since the key contribution of this method combines (1) discrete prompt tokens and (2) ICL.

**Questions:**

1. My major concern is the main loss function. Can you clarify the rationale behind the main loss function minimizing the distance between the in-context predictions and the zero-shot predictions in Eq. (1)? If this loss is truly optimized, the trivial solution would just encourage the model to ignore the $z$ and $D_l$. That is, the in-context prediction with the learned prompt z, i.e., $f(x_l, z, D_l)$, would approximate the zero-shot performance without the learned prompt, i.e., $f(x_I, \emptyset, \emptyset)$. Then what benefits can the model have from learning $z$ and selecting $D_l$?  I would suggest a better loss function (for example, preference learning loss like DPO), where the model would prefer the outcome of $f(x_l, z, D_l)$ over the outcome of $f(x_I, \emptyset, \emptyset)$, therefore you encourage the model to leverage $z$ and $D_l$ to learn a better output.

2. Why is the entropy term necessary? More explanations should be added on Page 5, Line 258-260. The ablation of the entropy term only on SST-2 in Fig 3 (b) is not convincing. I'd suggest you perform more ablation experiments on other datasets (with an unsaturated performance of direct prompting). Moreover, different $\alpha$ will also influence the accuracy. Can you clarify your strategy for selecting $\alpha$ to balance the trade-off of accuracy and variance? I'd suggest you perform ablation studies on the effect of $\alpha$?

---

### Official Review · Reviewer_HmbH · 2024-11-05

**Soundness:** 2
**Presentation:** 2
**Contribution:** 2
**Rating:** 3
**Confidence:** 4

**Summary:**

The paper introduces an approach to unsupervised prompt learning for classification tasks using black-box large language models (LLMs), named as Prompt learning with Pseudo-labeled Demonstrations (PPD). It proposes to generate pseudo-labels from unlabeled data and using these as in-context demonstrations to learn prompts effectively. The authors claim that their approach can be used to improve performance on downstream tasks. The evaluations were conducted on several public benchmarks to demonstrate the effectiveness of the proposed method.

**Strengths:**

1. The paper is well-written and easy to follow.
2. The motivation to use pseudo-labeled data as demonstrations is clear and reasonable to understand.
3. The experiments are relatively extensive on several benchmarks.

**Weaknesses:**

1. The novelty of the proposed method is very limited. The idea of using unlabeled data as pseudo labels is very intuitive and covered in many previous papers, including but not limited to [1,2,3,4]. Even though the method of selecting pseudo-labeled samples is slightly different in these papers, the contributions are relatively marginal.
2. The proposed PPD to use VR-PGE optimization is complex for classification tasks. I would like to see if there are complexity analyses compared to other baselines.
3. The performance showed in the experiments are not convincing for such overly complicated method. I don't see much improvement of PPD even compared with vanilla ICL baselines. So I doubt the real efficiency and effectiveness of the proposed method in real practices.

[1] Abburi, Harika, et al. "Generative ai text classification using ensemble llm approaches." arXiv preprint arXiv:2309.07755 (2023).\
[2] Zhang, Yunyi, et al. "Teleclass: Taxonomy enrichment and llm-enhanced hierarchical text classification with minimal supervision." arXiv preprint arXiv:2403.00165 (2024).\
[3] Zhang, Yunyi, et al. "PIEClass: Weakly-supervised text classification with prompting and noise-robust iterative ensemble training." arXiv preprint arXiv:2305.13723 (2023).\
[4] Mirza, Muhammad Jehanzeb, et al. "Lafter: Label-free tuning of zero-shot classifier using language and unlabeled image collections." Advances in Neural Information Processing Systems 36 (2024).

**Questions:**

See the weaknesses above.

---

### Note · Authors · 2024-11-19

I have read and agree with the venue's withdrawal policy on behalf of myself and my co-authors.